# Porcine Nose Atrophy Assessed by Automatic Imaging and Detection of *Bordetella bronchiseptica* and Other Respiratory Pathogens in Lung and Nose

**DOI:** 10.3390/ani14213113

**Published:** 2024-10-29

**Authors:** Hanna Lichterfeld, Sara Trittmacher, Kathrin Gerdes, Kathrin Schmies, Joaquín Miguel, Irene Galé, Alba Puigredon Fontanet, Isaac Ballarà, Krista Marie Tenbrink, Isabel Hennig-Pauka

**Affiliations:** 1Field Station for Epidemiology, University of Veterinary Medicine Hannover, Foundation, 49456 Bakum, Germany; hanna.lichterfeld@web.de (H.L.); sara.trittmacher@gmx.de (S.T.); kathrin.gerdes@tiho-hannover.de (K.G.); info-bakum@tiho-hannover.de (K.S.); 2HIPRA, 17170 Amer, Spain; joaquin.miguel@hipra.com (J.M.); irene.gale@hipra.com (I.G.); alba.puigredon@hipra.com (A.P.F.); isaac.ballara@hipra.com (I.B.); 3HIPRA Deutschland GmbH, 40211 Düsseldorf, Germany; krista.tenbrink@hipra.com

**Keywords:** lung health, nasal lesion score, nose deformities, respiratory disease, swine, *Bordetella bronchiseptica*, porcine respiratory disease complex

## Abstract

In this study, nursery pigs and those from the beginning of the fattening period with respiratory disorders were examined for nasal conchae atrophy by visual and automated scoring. Detection rates of *Bordetella (B.) bronchiseptica* in the nose and in the lung were related to the determined nasal lesion scores. In dead pigs submitted for necropsy, a routine bacteriological examination of the lower respiratory tract was performed. Moreover, nasal cross sections were scored by visual inspection and by automatic imaging based on artificial intelligence using the application AI DIAGNOS (HIPRA). No *Pasteurella* (*P*.) *multocida* positive for the *P. multocida* toxin were found so all farms were negative for progressive atrophic rhinitis. The findings suggest that *B. bronchiseptica* in the nose can facilitate lung infections, as a higher number of pathogenic bacterial species was found when *B. bronchiseptica* was present in the nose. However, no association was found in this study between *B*. *bronchiseptica* and non-toxigenic *P. multocida* and nasal alterations. Nose health might be decisive for respiratory health. Study results highlight the nose as an important reservoir for *B*. *bronchiseptica* and *P. multocida*.

## 1. Introduction

Multifactorial respiratory disease in swine is characterized by farm-specific combinations of various biotic and abiotic factors including pathogens of major or minor importance [1,2,3,4,5].

Some authors classified the gram-negative bacterium *Bordetella* (*B*.) *bronchiseptica* as a major pathogen together with porcine reproductive and respiratory syndrome virus (PRRSV), *Actinobacillus* (*A.*) *pleuropneumoniae*, *Mesomycoplasma* (*M.*) *hyopneumoniae*, swine influenza virus (SIV), pseudorabies virus, *Actinobacillus (A.) suis*, and *Salmonella* spp. [3], while others addressed the porcine respiratory corona virus (PRCV), PRRSV, SIV, *A. pleuropneumoniae*, porcine circovirus (PCV) 2, and *M. hyopneumoniae* as primary pathogens [4]. Primary pathogens can induce severe lung lesions just by their virulence properties in contrast to opportunistic or secondary pathogens, which induce lung lesions if accompanied by other co-infecting pathogens, harmful conditions, or immunosuppressive factors. Secondary pathogens are often commensal colonizing bacteria in the upper respiratory tract, especially in the nasal cavities or on the tonsils [3]. One reason why *B. bronchiseptica* is regarded as a primary pathogen by some authors is that the dermonecrotic toxin causes pneumonic lesions and nasal conchae atrophy accompanied by necrosis, accumulation of inflammatory cells (neutrophils and macrophages), and bronchiolar epithelial hypertrophy especially in young pigs [6]. Clinical symptoms of *B. bronchiseptica* infections mainly include sneezing and coughing. Coinfections with *B. bronchiseptica* as a secondary pathogen mostly cause more severe courses of disease than the monoinfection with the primary pathogen. Mechanisms of infection and synergisms with other pathogens have been elucidated in in vitro models [7,8,9,10].

In nursery and fattening pigs, *B. bronchiseptica* can be an opportunistic pathogen contributing to respiratory disease, enhancing colonization with *Streptococcus* (*S.*) *suis* and *Glaesserella* (*G.*) *parasuis*, promoting disease caused by *S. suis*, and interacting with PRRSV and SIV to increase the severity of respiratory disease [5,7].

The first line of defense against airborne pathogens, particles, and gases causing respiratory disease is the nose with the function of air warming, filtering, and mucosal immune response [11]. In case of nasal conchae atrophy the filtering function is disturbed. The detection of nasal conchae atrophy and subsequent examination for toxigenic *P. multocida* expressing the *P. multocida* toxin (PMT) for exclusion of progressive atrophic rhinitis (PAR) is an important diagnostic step. Examination for *B. bronchiseptica* as a cause for non progressive atrophic rhinitis (NPAR) is also of high importance. The nasal turbinate deformations in NPAR are reversible and milder in contrast to those in PAR [12]. For assessment of nasal conchae atrophy in pigs in nasal cross sections a standardized scoring scheme exists [13,14], which was a good starting point for development of an automatic assessment tool using artificial intelligence (AI). The assessment of organ alterations with the help of AI in healthcare is still in its early stages also in human medicine. In so called supervised learning algorithms deep learning models are trained on vast datasets of medical images, which are annotated by medical experts to indicate the presence and severity of specific organ changes [15]. By training the deep neuronal network with these annotated images, the AI models can learn to identify patterns and features associated with different conditions. When presented with a new image, a successfully trained network can then provide a probability or classification of the organ’s health status. The acquisition of well-annotated datasets is the most challenging part of training AI models. The selection of divers and broad data for training is crucial to avoid overfitting of the AI models and help them to generalize well on other datasets or populations [16].

So far most promising current applications in humans are cancer detection in histopathology in various organs such as breast, lung and skin, as well as automation of radiology and computer tomography reports on organ shape, size and texture [17].

In livestock medicine recent digital advances in sensor technology drive research on implementation of AI for livestock monitoring with regard to detection of changes from physiological conditions related to different stressors and early prediction of disease [18]. In contrast to that the assessment of macroscopic organ alterations by AI in veterinary medicine is not implemented in practice so far. In this study the assessment of images of porcine nasal cross sections was performed using the application (app) Artificial Intelligence Diagnos (AI Diagnos, HIPRA, Amer, Spain) as a fully automated diagnostic system available for ios and android, which was developed to simplify, objectify and facilitate the whole process of slaughterhouse assessment [19].

In this study an automatic scoring of images of nasal cross sections by the app AI DIAGNOS was compared to a visual scoring of cross sections of nasal turbinates by a specialist to assess nasal conchae atrophy in pigs with respiratory disease [19]. The further aim of the study was to assess the impact of *B. bronchiseptica* in the upper and lower respiratory tract as a predisposing factor of nasal lesions and coinfections, respectively.

## 2. Materials and Methods

### 2.1. Sample Collection

In total, 121 pigs from 64 different farms with respiratory disease problems were included in the study within two periods covering a total of 11 sampling months (August 2022 to January 2023 and October 2023 to February 2024). Pigs were selected for further diagnostics due to coughing and were sent for diagnostic necropsy to the Field Station for Epidemiology of the University of Veterinary Medicine Hannover, Foundation, Hannover, Germany. Body weights of the pigs ranged from 3.7 kg to 42 kg with an average weight of 15.6 kg. The age of the pigs could not be deduced from the body weights because most of the pigs might have wasted according to inflammatory lung alterations, which were detected in all pigs at necropsy. All pigs originated either from nursery or early fattening units.

A routine bacteriological examination of altered lung tissue of a lobar bronchus and the lung periphery was performed in the microbiology laboratory of the Field Station for Epidemiology of the University of Veterinary Medicine Hannover. Focus of the microbiological examinations of the lower respiratory tract was on the pathogens *B. bronchiseptica*, *P. multocida*, *G. parasuis*, *S. suis*, *A. pleuropneumoniae*, ß-hemolytic streptococci, and *Escherichia (E.) coli* (Table 1).

The upper respiratory tract with focus on the nose was inspected in parallel by two different techniques, an AI tool (AI DIAGNOS) and visual inspection [19].

After separating the head from the body, one ear was fixed with one hand so that the scalp could be removed with a sharp knife starting from the cut edge on the neck. Skin removal on the nose and upper jaw was the precondition for saw positioning with sawn teeth already having an abutment in the bone. At the height of the first premolar tooth in the upper jaw, where the dorsal and ventral conchae are most pronounced in healthy pigs, the nose was cross-sectioned with an electric band saw [20]. Two sterile culture swabs without transport medium (Culture swab without transport medium, Nerbe plus GmbH & Co. KG, Winsen/Luhe, Germany) were used to sample the left and right ventral conchae of the head-sided cross-section for further bacteriological and molecular biological examination (Table 1) using each swab for both sides. The first nasal swab was used in parallel with the lung tissue samples for cultural microbiological examination following routine methods of the DIN EN ISO 17025 [21] accredited lab. The bacteriological examination of the nasal swab focused only on detection of *B. bronchiseptica* and *P. multocida*. The second nasal swab was cryopreserved at −80 °C (Cryobank, Mast Group Ltd., Bootle, UK) until shipment in a polystyrene box with cold packs for PCR diagnostic for *B. bronchiseptica* by an in-house method at the HIPRA laboratories in Amer, Girona, Spain [22]. Nasal swab diagnostic only aimed at detecting *B. bronchiseptica* and *Pasteurella multocida* and no other bacterial or viral pathogens.

The nasal cross section was cleaned by flushing with tap water, and photographed with a single lens reflex camera (DMC-FZ1000, Panasonic Holdings Corporation, Kadoma, Japan). Afterwards, the nasal cross section was scored visually following a standardized scoring scheme by the same experienced observer [13,14]. Automated scoring based on the photos of the nasal cross sections was performed retrospectively using the AI DIAGNOS app (version 3.3.0).

Metadata for every animal included in the study as weight and the farm of origin were assigned to the visual and automatically generated scores as well as to the lab results.

### 2.2. Bacteriological Examination

For bacteriological examination, nasal swabs and lung tissue samples were fractionally smeared on chocolate blood agar (NAD, Blood Agar no. 2, Becton, Dickinson and Company, Sparks, NV, USA), CNA blood agar (Becton, Dickinson and Company), Gassner agar (Oxoid Ltd., Hampshire, UK) and Columbia agar with 5% sheep blood (Becton, Dickinson and Company). Incubation of inoculated agar plates (Colombia, Gassner, CNA) took place at 36 °C for 48 h under a standard atmosphere. An 8% CO_2_ atmosphere was used for the incubation of the chocolate blood agar. The plates were examined twice, after a 24-h and a 48-h incubation period. Further characterization of colonies was based on biochemical characteristics. Colonies that looked like *B. bronchiseptica* (smooth, curved, and translucent) were tested for oxidase and urease. Colonies similar to *P. multocida* (smooth, whitish, with indole-like smell/odor) were tested for catalase and indole. In the case of positive results for *P. multocida*, a modified real-time PCR was performed to detect the gene for the dermonecrotic toxin of *P. multocida* (*toxA* gene) [23]. *G. parasuis*- and *A. pleuropneumoniae*-like colonies (dewdrop-shaped, cream-colored) were tested for catalase and for the CAMP phenomenon. Colonies similar to *Streptococcus* ssp. (smooth, flat, and translucent to whitish with hemolysis) were tested for starch. Colonies typical of *E. coli* (grayish, wet, hemolytic) were tested using lactose.

All bacteriological examinations followed routine procedures of the accredited diagnostic lab.

### 2.3. Molecular Biological Detection by PCR

The extraction of *B. bronchiseptica* DNA was performed directly in accordance with the HIPRA DIAGNOS laboratory in-house extraction protocol for nasal swabs.

A real-time *B. bronchiseptica* PCR targeting a 237 bp amplicon of the FlaA (flagellin) gene was performed. This in-house PCR was developed and validated by HIPRA DIAGNOS laboratory in 2013. The cut-off of this PCR for *B. bronchiseptica* detection was at a cycle threshold (ct) of 38.5, using a modified method based on the *B. bronchiseptica* PCR described by Hozbora et al. (1999) where the reaction mixture contained 50 mM KCl, 10 mM Tris hydrochloride (pH 8.4), 2.5 mM MgCl2, 1 mg gelatin per mL, 400 µM deoxyribonucleotide, 1 µM primer, and 0.5 U of Taq polymerase (Beckman Coulter, Inc., Fullerton, CA USA). Amplification took place in 35 cycles with 1 min of denaturation at 94 °C, 15 s of annealing at 53 °C, and 20 s at 72 °C [22].

The first step of DNA extraction from colony material of *P. multocida* from pure cultures was the suspension of colony material in 500 µL Tris-EDTA buffer (TE Buffer (1×) pH 8.0 for molecular biology, Apollo Scientific Ltd., Stockport, UK), followed by centrifugation for 10 min at 20,000× *g* (Centrifuge 5424, Eppendorf SE, Hamburg, Germany). The supernatant was discarded, and the pellet was resuspended by adding 500 µL of Lysis Buffer Tris-HCL pH 8.5 in an Eppendorf Cup 2 mL (Safe-Lock Tubes 2.0 mL, Eppendorf SE). The samples were then shaken on a vortex, incubated for one hour at 60 °C, and for 15 min at 95 °C in a thermal shaker (ThermoMixer C, Eppendorf SE). After centrifugation for 10 s at 6000× *g*, the supernatant with the DNA extract was collected.

A DNA fragment coding for a part of the *P. multocida* toxin (PMT) was targeted by a modified in-house real-time duplex PCR based on a method described by Scherrer et al. [23]. The master mix consisted of 6.25 µL ABI Universal Master Mix (TaqMan^®^ Universal Master Mix, Thermo Fisher Scientific, Inc., Waltham, MA, USA), 0.25 µL primer probe mix (primer F and R concentrations 400 nM each (Metabion international AG, Planegg, Germany), probe concentration of 200 nM (Metabion international AG)), 0.4 µL ABI IPC Master Mix, 0.1 µL ABI IPC Template (TaqManTM Exogenous Internal Positive Control Reagents (VIC^TM^ Probe, Thermo Fisher Scientific, Inc.)), and 4.25 µL H_2_O. To this, 1.25 µL DNA template was added. The amplification protocol was 10 min at 95 °C and then 40 cycles for 15 s each at 95 °C and for 60 s at 60 °C.

### 2.4. Nasal Lesion Scoring (NLS)

Nasal turbinates were scored visually in nasal cross sections during necropsy and retrospectively by an independent scoring of the respective digital image of the nasal cross section using the AI software in form of a user friendly application for a mobile phone (AI DIAGNOS version 3.3.0). In this study, relevant nasal conchae atrophy was defined from a score of ≥5 (moderate: NLS 5–8, severe: NLS 9–18) as published elsewhere [14].

Visual scoring was performed separately on each of the four scrolls of the ventral nasal turbinates s and the nasal septum according to the criteria shown in Table 2.

Combining all five scores in a summation score, the maximum reachable score was 18 [13]. Examples for different visual scores recorded by a skilled observer are shown in Figure 1a, Figure 2a, Figure 3a, Figure 4a and Figure 5a.

The app AI DIAGNOS is a computer vision system (CVS) developed by HIPRA. It is used to recognize and score nasal lesions related to atrophic rhinitis on digital images captured from noses of dead pigs. The CVS was created by using Amazon SageMakerTM^®^ (Amazon Linux 2 and Jupyter Lab 3 (notebook-al2-v2)), which is an everything-as-a-service cloud machine learning platform from Amazon Web Services. SageMakerTM^®^ offers pre-built algorithms and frameworks specifically designed for CVS tasks like image classification and object detection and enables developers to create, train and deploy own machine learning models in the cloud. To detect and classify the relevant parts of the images the following steps were implemented into the model pipeline (i) preprocessing by removing noise and adjusting contrast from the image, (ii) identifying and extracting relevant features from the images, such as edges, corners, or textures, (iii) object identification within the image, (iv) image segmentation into different regions and (v) interpretation by translation into scores [19]. In a first step, a nose image passed through a filter (focus detector) with the aim of differentiating the nose from the background. Subsequently, the processed image was passed through an area of interest detector that identified each nasal scroll and the central septum. These objects were passed through the final convolutional neuronal network which was trained to predict the nasal lesion score based on the European Pharmacopeia guidelines [13].

The system was trained using 5975 images photographed with a smartphone camera. These images were manually annotated with a target score by five experts from different countries based on the European Pharmacopea guidelines for atrophic rhinitis and fed into the model as a training set [13]. This procedure enabled the trained model to automatically recognize the different tissue areas in the uploaded pictures (examples in Figure 1b, Figure 2b, Figure 3b, Figure 4b and Figure 5b) and to score the respective lesions for each area in the image based on the same scoring scheme used for visual scoring. The trained model prototype of the app was provided by the manufacturer for validation in the first diagnostic field application in this study. Retrospectively, the app produced scores were compared to the visual scores.

Figure 1, Figure 2, Figure 3, Figure 4 and Figure 5: Photos of the nasal cross sections from five of the 121 examined pigs; (**a**) visual scoring (**b**) automated scoring; boxes in (**b**) = regions of interest created by the app for scoring; vl = ventral left; dl = dorsal left; vr = ventral right; dr = dorsal right; S = nasal septum. The numbers next to the evaluated anatomical parts represent the respective scores for this part. The total nasal lesion score (NLS) defined either visually (**a**) or automatically by the app (**b**) is depicted beyond the respective photo.

### 2.5. Data Management and Statistical Evaluation

All data were recorded in Excel, version 2016 (Microsoft Corporation, Albuquerque, NM, USA). Data analyses were performed with statistical software R (version 4.3.1, R Core Team, 2023). Associations between positive findings for *B. bronchiseptica* and nasal scores were analyzed by linear regressions. Ordinal and not normally distributed data were compared by the Mann-Whitney U test. The correlation between visual and automated NLS was tested by Spearman’s rank correlation tests. Proportions in different groups were compared by the Fisher’s exact test. The level of significance for all statistical models was *p* = 0.05.

## 3. Results

### 3.1. Pathological and Bacteriological Findings

Piglet body weights were in the range of 3.7 kg to 42 kg (mean ± standard deviation: 15.6 ± 9.1 kg), while information about the age was not available.

Due to bacterial overgrowth, 14 nasal swab samples could not be evaluated by bacteriological culture, while PCR was successful with all samples. *B. bronchiseptica* was detected in 66 of 121 nasal swabs (54.5%) by PCR and in 25/107 (23.4%) by bacteriological examination (Table 3). In total, pigs from 64 farms were examined and pigs from 39 farms were tested positive for *B. bronchiseptica* by PCR in the nose (60.9% positive farms). Parallel bacteriological culture of nasal swabs for *P. multocida* resulted in 29.0% (31/107) positive pigs. Of pigs positive by PCR for *B. bronchiseptica* in nasal swabs, 26.3% (15/57) were also positive for *P. multocida*. In total, 32% of the pigs negative for *B. bronchiseptica* in nasal swabs were positive for *P. multocida* (16/50). All *P. multocida* isolates were negative for the PMT.

In total, 121 swabs were tested by PCR, while bacterial culturing was only possible in 107 swabs due to overgrowth with unspecific bacterial flora. The cut-off cycle threshold (ct) for a positive PCR result of *B. bronchiseptica* was ct < 38.5.

In 109/121 pigs (90.1%) a catarrhal purulent bronchopneumonia, in six pigs (5.0%) a purulent pneumonia with lung abscesses, and in six pigs (5%) a hemorrhagic necrotizing pneumonia were diagnosed. Pleuritis characterized by fibrinous plaques on the serosal surface, or pleural adhesions was detected in 51/121 of the pigs (42.1%).

In 30/109 pigs (27.5%) with catarrhalic purulent bronchopneumonia, no pathogenic bacteria were isolated.

Independent of the character of pneumonia, the most common pathogens in the respiratory tract were *S. suis* (33/121, 27.3%), *G. parasuis* (30/121, 24.8%), *B. bronchiseptica* (27/121, 22.3%), and *A. pleuropneumoniae* (13/121, 10.7%), as shown in Figure 6. Coinfecting agents in the respiratory tract were evaluated in lungs positive for *B. bronchiseptica* (*n* = 27), resulting in *S. suis* (6/27; 22.2%), *G. parasuis* (5/27; 18.5%), *A. pleuropneumoniae* (4/27; 14.8%), E. coli (3/27; 11.1%), and ß-hemolytic streptococci (2/27; 7.4%). In 10/121 (8.3%) pigs only, *B.bronchiseptica* was cultured from lung tissue.

A significantly higher number of a few specific pathogenic bacterial species in the lower respiratory tract was found in pigs positive for *B. bronchiseptica* in the nose (*p* = 0.005) (Figure 7).

Pigs PCR-positive for *B. bronchiseptica* in the nose were significantly more often positive for *B. bronchiseptica* in the lower respiratory tract (*p* < 0.001, Figure 8). Of 66 pigs PCR-positive for *B. bronchiseptica* in the nose, 25 pigs were positive for *B. bronchiseptica* in the lower respiratory tract (37.9%).

### 3.2. Association of Nasal Lesion Scores and B. bronchiseptica

No association of nasal lesion scores and detection rates for *B. bronchiseptica* by PCR in noses was found. In total, 43/121 pigs (35.5%) showed nasal conchae atrophy (NLS 5–18). In these pigs, 58.1% (25/43) were positive for *B. bronchiseptica* in the nose. No nasal alterations (NLS = 0) were detected in 8.3% (10/121) of the pigs, while mild alterations (NLS 1–4) were found in 56.2% of the pigs (68/121). In the group of pigs with no or mild nasal lesions (NLS ≤ 4), *B. bronchiseptica* was detected in 52.6% of the pigs (41/78). The difference in *B. bronchiseptica* detection rates in pigs with low (NLS ≤ 4) and high NLS (5–18) was not significant (*p* = 0.95) (Figure 9).

### 3.3. Comparison of Visual Scores for Nasal Turbinate Alterations with Automated Scores Generated by AI DIAGNOS

All 121 nasal cross sections received one visual and one automated score between 0 and 18. The automated NLS were in the range of 0–12 (median 3), and the visual NLS in the range of 0–15 (median 4). The automated scores were significantly correlated with the visual scores (Spearman’s rank correlation coefficient *r_s_* = 0.61; *p* < 0.001), as shown in Figure 10.

## 4. Discussion

*B. bronchiseptica* can either be isolated in the upper respiratory tract of healthy pigs [24] or can cause specific disease pattern [25]. Translocation of *B. bronchiseptica* colonizing the nasal mucosa into the lower respiratory tract can be assumed.

In this study, pigs from 25 farms were negatively tested for *B. bronchiseptica* in the nose. The sample size of pigs per farm was not adequate to state that a farm was free of *B. bronchiseptica*. Due to the fact that *B. bronchiseptica* is a nasal colonizer in healthy pigs, it can be assumed that all farms were positive.

*B. bronchiseptica* and *P. multocida* can colonize healthy pigs without causing rhinitis [24], but *B. bronchiseptica* in particular can be a precursor for other pathogens and, as our study shows, potentiate the presence of other pathogens (Figure 7). A significantly higher number of pathogenic bacterial species in the lower respiratory tract was found in pigs positive for *B. bronchiseptica* in the nose. This supports previously published observations that *B. bronchiseptica* in the upper respiratory tract can disrupt the nasal microbiome and lead to an increased abundance of other bacteria, especially with potential contribution to respiratory disease [26].

*S. suis*, *G. parasuis*, *B. bronchiseptica*, and *A. pleuropneumoniae* were most frequently detected bacterial pathogens in lung tissue of pigs with respiratory disease in this study. Synergistic effects between *B. bronchiseptica* and most of these bacteria have been previously described [7,27,28]. Further interactions between *B. bronchiseptica* and other bacterial pathogens as e.g., *Mesomycoplasma (M.) hyorhinis* and viral pathogens as SIV or PRCV have been published previously but were not investigated in this study [9,10,29,30,31].

Unexspectedly more than one third of pigs showed nasal lesions in this study, although all *P. multocida* isolates lacked the *toxA* gene and no correlation of *B. bronchiseptica* and *P. multocida* with macroscopic nasal lesions was found (Figure 9).

In previous studies using the same scoring scheme, nasal atrophy was related to weight and daily weight gain. Pigs with moderate (5–8) and severe (9–18) NLS showed reduced growth rates and finally lower slaughter weights than healthy pigs with NLS 0–4 [14]. In contrast to our study, in the cited study the examined pigs were suffering from PAR due to infection with *B. bronchiseptica* and toxigenic *P. multocida* strains with PMT. In our study, examined pigs with NLS ≥ 5 were negative for the PMT and therefore NPAR was diagnosed.

The question whether NPAR also leads to reduced growth could not be answered in our study because one weak point were the lacking metadata on age of the examined pigs. Farmers sent pigs for necropsy without reporting or even knowing the correct age of the animals. Underweight pigs were often kept longer in a respective unit and housed with pigs of younger age-groups. Finally, a relation between the severity of the NLS to the age of the examined pigs was not possible, as only the weight was known. Under field conditions with various biotic and abiotic factors, the process of nasal turbinate atrophy might start after weaning in the immunological gap when maternal antibodies have decreased and active humoral immune responses are still underdeveloped [32,33]. Nasal mucosal inflammation is reported to be more common in young pigs [32], possibly because the epithelial and immunological barrier of the nasal mucosa as well as the protective microbiota are less developed than in older pigs [34].

A second weak point of this study is that histological examination of the nasal mucosa as the gold diagnostic standard was not performed. In addition to histological evidence of rhinitis, an involvement of cytomegalovirus could have been diagnosed by pathognomonic intranuclear inclusion bodies [35]. On the other hand, interpretation of mostly unspecific histological findings in mucosal sites with an interface with the environment is difficult, which was shown for porcine tonsils with immune cell aggregates and bacterial colonization already in healthy pigs [36]. It can be assumed that histological examination of the nose in this study would have meant no substantial gain in information because nasal conchae atrophy was already visible macroscopically. An exception is the lacking diagnostic for cytomegalovirus, which could have been done histologically but also by PCR.

This reveals the third weak point of this study; no bacteria other than *B. bronchiseptica* and *P. multocida* and no viral pathogens were targeted in the diagnostic from the nasal swabs. The diagnostic of other pathogens potentially involved in atrophy in the nasal conchae and the combination of pathogens would have been of interest. On the other hand, several recent studies of the nasal microbiota revealed that not only the detection of single pathogenic species but also the microbiota might be decisive for nasal alterations [34].

Different methods for visual scoring of the nasal turbinates and quantification of the grade of atrophy have been elaborated, which are all based on subjective assessment [37,38]. Therefore, results of nasal scorings performed by different observers should be compared with caution [39].

In order to increase objectivity in the analysis of the nasal cross sections and to enable permanent availability of all past evaluations of the images, a new app prototype based on AI algorithms was tested in this study (AI DIAGNOS). A summary of the user experience gained in this study is as follows: (i) the smart phone photo quality must be adequate (sharpness, lightness, distance), which means that photos should have at least 5–6 megapixels, which is in the range of the last five conventional smartphone generations [40], (ii) the cut to produce the nasal cross section must follow a perpendicular line, (iii) the nasal cross section is not frayed and blood- or mucus free. These preconditions mean additional work load for the observer before using the app. As shown in Figure 3b and Figure 5b, boxes of interest created by the app do not always correspond to the anatomical region of interest. In Figure 3b, the clearly defined dorsal recessus of the conchae in an intense reddish color were selected by the app instead of the dorsal conchae, and a wrong score of 0 was assigned. In Figure 5b, complete atrophy of the ventral turbinates of the ventral conchae is shown and the app was unable to determine the correct region of interest due to lacking anatomical structures. As a result, both frames on the left side overlapped each other, leading to false automatic scoring. The degree of correlation between visual and automated NLS reflected by the Spearman rank correlation coefficient (*r_s_* = 0.61), as shown in Figure 10 is negatively influenced by outlier NLS pairs due to incorrectly placed frames during automated scoring. This resulted in high deviations between both scores. This situation can be deduced from visualized results in Figure 10, where automated scoring resulted in NLS 0 and visual scoring in a high score (e.g., NLS 9). In most NLS pairs, only slight deviations were observed so the Spearman rank correlation coefficient still reflects a high correlation between both scores.

The future goal is that app-based automatic scoring becomes less biased among different observers. The observers can already benefit from this advantage if the frames are moved manually to the correct positions prior to evaluation. This app is already available and can already be used on a large scale so herds can be classified according to nasal health checks at necropsy or the slaughterhouse.

## 5. Conclusions

Assessment of nasal conchae atrophy by a new app based on artificial intelligence used in this study was successful in comparison to visual scoring. To avoid misplacement of the frame to the region of interest by the app, further model learnig on object identification and image segmentation should be performed. In this study, a high percentage of pigs (35.5%) with respiratory symptoms showed atrophy of the nasal conchae. Detection rates of *B. bronchiseptica* and *P. multocida* were not related to the nasal lesion scores, so that their impact as predisposing factors could not be shown in this study. *B. bronchiseptica* in the nose was associated with the presence of *B. bronchiseptica* in the lower respiratory tract and also with the number of pathogenic bacterial species in the lung. We can conclude that *B. bronchiseptica* increased the coinfections in the examined pigs and that nasal health has a major impact on respiratory health in general.

## Figures and Tables

**Figure 1 animals-14-03113-f001:**
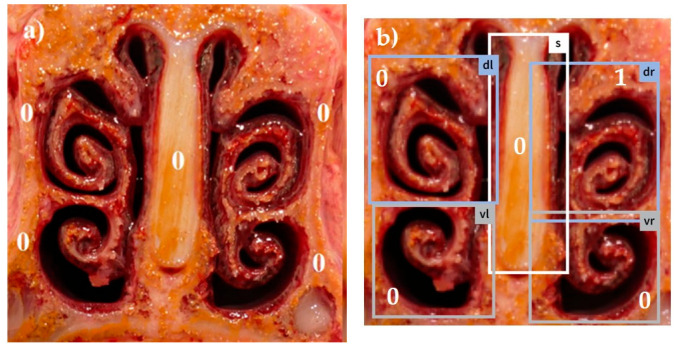
(**a**) NLS 0; (**b**) NLS 1.

**Figure 2 animals-14-03113-f002:**
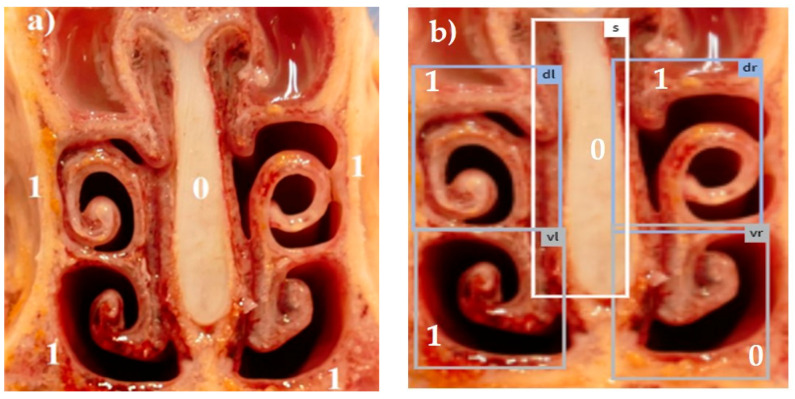
(**a**) NLS 4; (**b**) NLS 3.

**Figure 3 animals-14-03113-f003:**
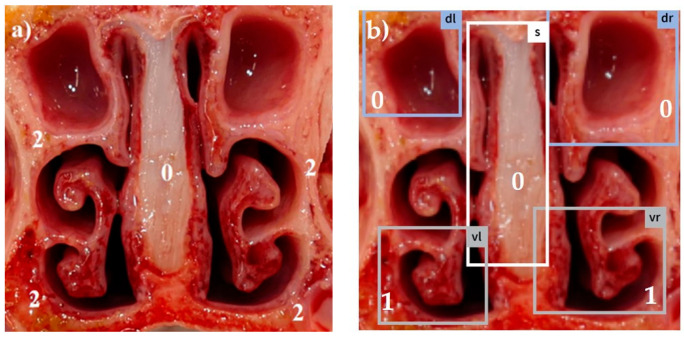
(**a**) NLS 8; (**b**) NLS 2.

**Figure 4 animals-14-03113-f004:**
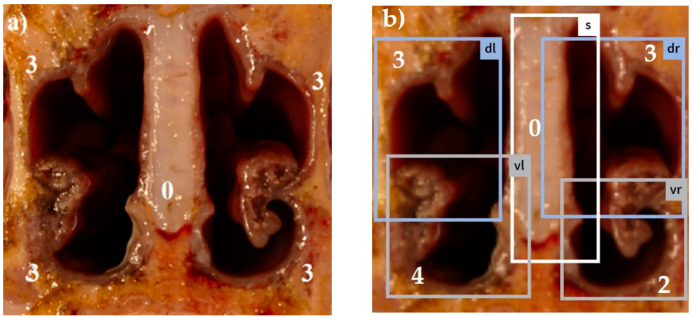
(**a**) NLS 12; (**b**) NLS 12.

**Figure 5 animals-14-03113-f005:**
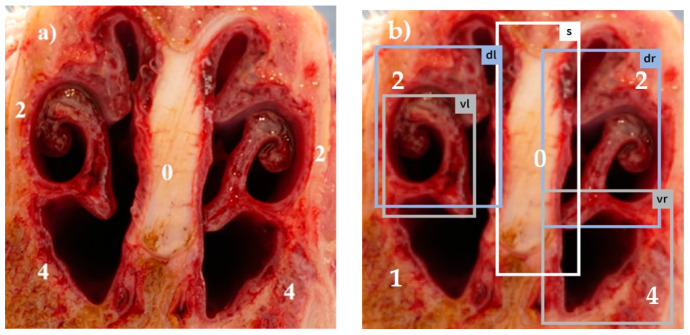
(**a**) NLS 12; (**b**) NLS 9.

**Figure 6 animals-14-03113-f006:**
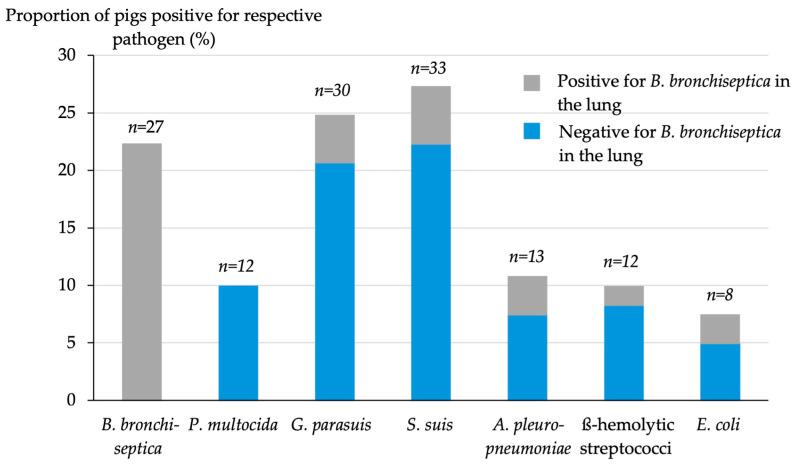
Prevalences of different pathogens in the lower respiratory tract (*n* = 121); numbers above the bars represent the number of pigs positive for the respective pathogen. The shaded grey parts of the bars depict the proportion of pigs simultaneously positive for *B. bronchiseptica* in the lung.

**Figure 7 animals-14-03113-f007:**
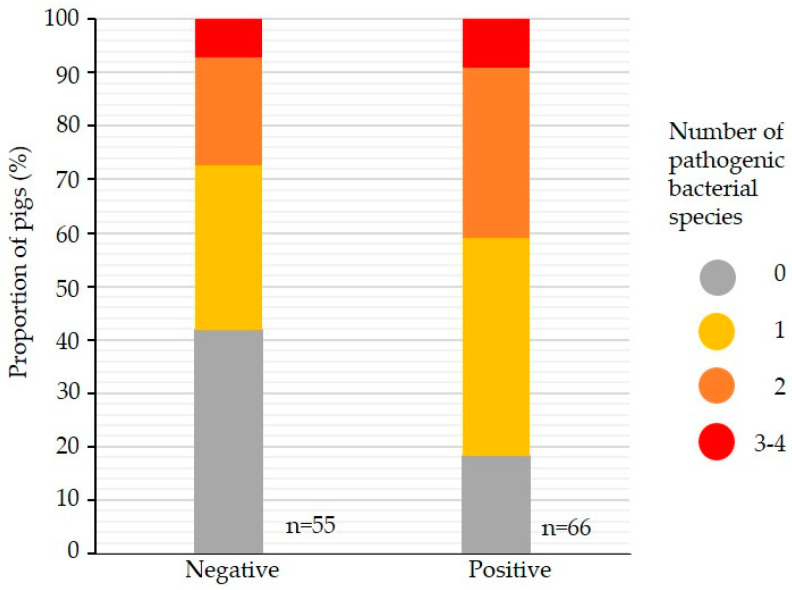
Number of pathogenic bacterial species in the lower respiratory tract in pigs positive and negative for *B. bronchiseptica* in the nose by PCR (*n* = 121).

**Figure 8 animals-14-03113-f008:**
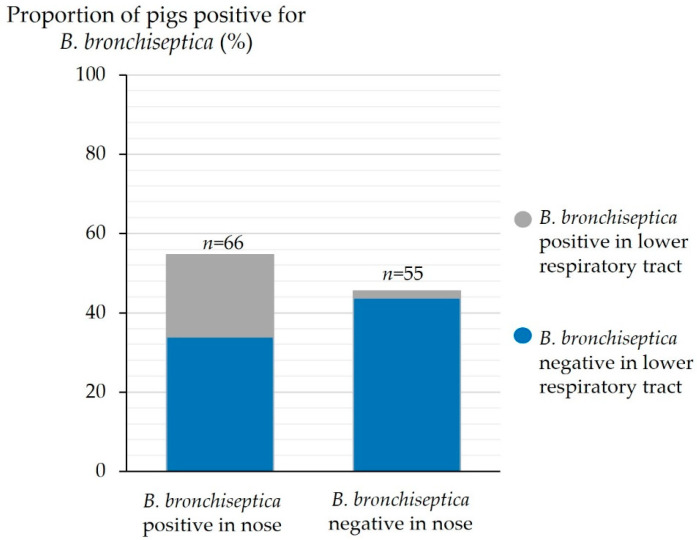
Detection of *B. bronchiseptica* in the nose and the lower respiratory tract (*n* = 121).

**Figure 9 animals-14-03113-f009:**
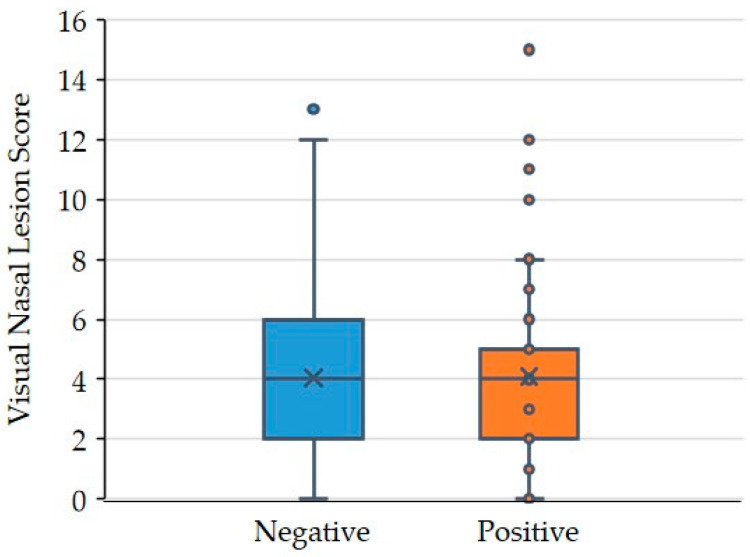
Association between *B. bronchiseptica* and severity of nasal turbinate lesions (*n* = 121); blue (negative for *B. bronchiseptica* in nose) and orange boxes (positive for *B. bronchiseptica* in nose) represent the interquartile data of the nasal lesion score (50% between 25% and 75% quartiles). The lines inside the boxes indicate the median. The upper and lower fences are defined as first and third quartile (represented by the lower/upper edge of the box) with minus/plus 1.5 times the interquartile range (IQR) indicating outliers (dots outside the boxes).

**Figure 10 animals-14-03113-f010:**
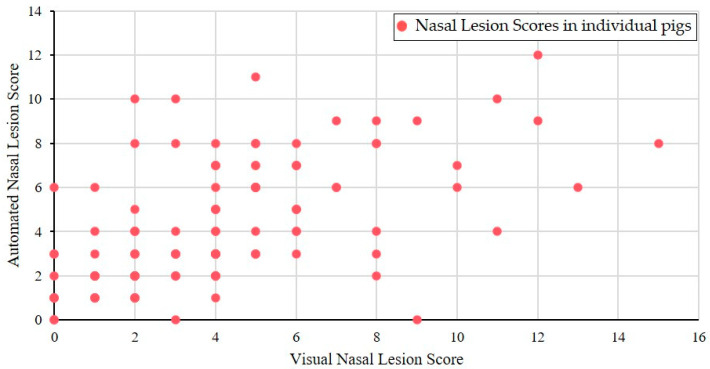
Correlation between visual and automated nasal lesion scores (*n* = 121).

**Table 1 animals-14-03113-t001:** Inspected pathogens in this study.

Obtainment Site	Assay	
	Culture and Isolation	PCR
Nose	*B. bronchiseptica* *P. multocida*	*B. bronchiseptica* *P. multocida* (*toxA* gene)
Lower respiratory tract	*B. bronchiseptica* *P. multocida* *S. suis* *A. pleuropneumoniae* *G. parasuis* ß-hemolytic streptococci *E. coli*	*-*

**Table 2 animals-14-03113-t002:** Description of the Nasal Lesion Score, which is based on the Turbinate Atrophy Score and the Nasal Septum Deviation Score.

Score	Evaluation	Macroscopic Abnormality of the Nose
Turbinate Atrophy Score:		
0	No atrophy	Scrolls completely present (Figure 1a,b)
1	Mild atrophy	Absence of less than half of scrolls (Figure 2a,b)
2	Moderate atrophy	Absence of more than half of scrolls(Figure 3a,b)
3	Severe atrophy	Turbinate bone is straight without scrolls(Figure 4a,b)
4	Very severe atrophy	Complete absence of turbinate bone and scrolls(e.g., ventral scroll Figure 5a,b)
Nasal Septum Deviation Score		
0	Normal	No distortion of the nasal septum (Figure 1, Figure 2, Figure 3, Figure 4 and Figure 5)
1	Slight deviation	Slight deviation of the nasal septum
2	Severe deviation	Severe deviation of the nasal septum

**Table 3 animals-14-03113-t003:** *B. bronchiseptica* and *P. multocida* detection rates in nasal swabs.

	*B. bronchiseptica* PCR	
	Positive (*n* = 66)	Negative (*n* = 55)
**No culture possible due to overgrowth**	7.4% (9/121)	4.1% (5/121)
**Culture** **+** ***P. multocida* and *B. bronchiseptica***	7.5% (8/107)	-
**Culture** **+** ***B. bronchiseptica***	15.9% (17/107)	-
**Culture** **+** ***P. multocida***	6.5% (7/107)	15.0% (16/107)
**Culture** **−** ***B. bronchiseptica* and *P. multocida***	23.4% (25/107)	31.8% (34/107)

## Data Availability

All data generated and analyzed during this study are available from the corresponding author upon reasonable request.

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
