# Peer review of "Porcine Nose Atrophy Assessed by Automatic Imaging and Detection of Bordetella bronchiseptica and Other Respiratory Pathogens in Lung and Nose"

_animals, 2024, doi:10.3390/ani14213113_

Round 1
Reviewer 1 Report
Comments and Suggestions for Authors
Lichterfeld et al. presented a manuscript describing the presence of Bordetella bronchiseptica in the respiratory tract of necropsied pigs exhibiting respiratory symptoms, as well as the co-infection of B. bronchiseptica with other pathogens.
Although the topic is both interesting and relevant to the field of porcine health, the manuscript contains several issues that must be addressed and improved before it can be considered for publication.
Accordingly, please find below a list of problems that require revision:
In general
Overall, the manuscript is moderately well-written but has structural deficits, particularly in the results and discussion sections. Although I am not a native English speaker, I noticed that certain parts of the text could benefit from improvement in terms of language. I would kindly suggest having the manuscript reviewed by a native English speaker to enhance its clarity and readability.
Title
I would recommend including the use of the app in the title, as it represents the most innovative element of the article and, in my opinion, provides the most valuable results.
Change …by respiratory symptoms to for respiratory symptoms….
Abstract
Line 38: I noticed a discrepancy between the abstract and the results section; specifically, the abstract reports 29.9%, while the results section (line 261) mentions 28.9%. Please adjust.
Line 72: Please change the term “pneumonic” to “pneumonia” or “pneumonic lesions” and add the term “conchae” or “turbinate” after “nasal”. These modifications ensure clarity and precision.
Line 73: Reference #7 does not describe necrosis, neutrophils, and hemorrhages. It appears that the sentence has been copied from reference #3, which discusses the dermonecrotic toxin of Bordetella causing turbinate atrophy and pneumonic lesions characterized by necrosis, hemorrhage, neutrophil accumulation, and fibrosis. Please verify the references and ensure that they are cited correctly.
Line 77: Please specify what 'older pigs' refers to in the context of the study. Additionally, the sentence currently lacks a clear subject. It would be helpful to include 'B. bronchiseptica' as the subject to clarify the statement.
Material and Methods
Line 129: In addition to body weight, I recommend including the age of the animals in the study. Since all the examined pigs exhibited respiratory symptoms, I would expect that some might show signs of wasting. Furthermore, the described body weight range suggests that an animal weighing 3.7 kg in this age group could be wasting. The animals are described as nursery pigs; however, based on my experience, I have not encountered a nursery pig weighing 42 kg. Please provide additional information or confirm whether this animal was accurately classified as a nursery pig.
Line 130: Please specify the symptoms, indicating whether they affect only the upper respiratory tract (e.g., sneezing, coughing) or the lower respiratory tract (e.g., dyspnea), or if both are involved.
Figures 1b) – 5b): The photos in panel b) appear to be out of focus.
Results
Overall, the data are not clearly presented. It is confusing that the total number of animals varies, sometimes being reported as 122 and other times as 107. To enhance clarity, the sentence stating that 14 samples could not be cultured should be moved to the beginning of the paragraph rather than being placed at the end. Additionally, Figure 6 is not optimally designed; I recommend including the number of animals either next to or within the bars to improve clarity and provide a clearer representation of the data.
Line 271: Please add 'n=27' after 'lungs positive for B. bronchiseptica,' as this number is only visible in Figure 7 but not mentioned in the text. Additionally, specify the number of cultured pathogens (X/27; XX %) to make the information clearer and easier to read. It would also be beneficial to include the number of lungs with a monoinfection by B. bronchiseptica alone. According to my calculations, for 10 out of the 27 animals, it is unclear how many had E. coli cultured (as noted in Figure 7 but not discussed in the results or discussion sections), or if only B. bronchiseptica was cultured.
Line 283: Please clarify if B. bronchiseptica in the nose was detected by PCR or culture.
Figure 11: The label “Figure 11” is out of order, as is “Figure 10”, which comes afterward. This minor typographical error should be corrected.
Line 303 – 305: As previously mentioned in “Line 129”, I suggest using the age of the animals as a more appropriate parameter for statistical analysis or considering both parameters (age and weight) for the same reasons previously discussed. Including results based on these parameters could provide further insights.
Since the data are available, I would recommend including additional analysis and results.
Specifically, it would be interesting to illustrate which pathogenic bacteria from the lungs were cultured alongside multiple pathogens, particularly to see if the “co-infection” consistently involves the same 2 or 3 pathogens.
Additionally, in relation to the pattern of pneumonia described under 2.1 Sample Collection, it would be valuable to investigate whether there is a correlation between these patterns and the cultured bacteria. It is known which pathogens can lead to specific pattern of pneumonia. And such a correlation could be done at least for APP (hemorrhagic necrotizing pneumonia) and GPS (fibrinous pleuritis).
Discussion:
In general, the discussion could benefit from clearer structuring and a stronger focus on the results of this study. A more organized presentation would enhance the interpretation of the findings and their relevance.
Line 352 - 353: This sentence is unclear, particularly with the use of the word “affecting”, which does not fit in the context. Please revise it to improve clarity.
Line 353: Reference #45 Please verify the references and ensure that they are cited correctly.
Line 355: Reference #46 Please verify the references and ensure that they are cited correctly. The article does not provide information on S. suis causing atrophic rhinitis.
Line 358 – 361: The information in these sentences could have been confirmed or excluded in this study through histological examination. Such an investigation would have been appropriate in the study design to provide further results and validation of the findings.
Line 360: First use of an abbreviation should be written out in full terms. Please make the necessary changes.
Line 361: References #12,48,49: Please verify the references and ensure that they are cited correctly.
Line 365: Mycoplasma hyorhinis has been renamed Mesomycoplasma hyorhinis. Please update the taxonomic classification.
Line 373: Based on Reference #53, the cited article does not report that nasal mucosal inflammation is more common in young pigs. Please review the literature carefully and ensure that it is cited accurately.
Line 385: As mentioned previously, establishing a correlation between the NLS and the weight of the animal is inappropriate. Consequently, any conclusions regarding the NLS not being age-related should be reconsidered.
Line 387: The information regarding the average weight of the study animals being 13 gg should be included in either the M&M or results section.
Line 399: P. multocida typographical error, should be written in italics.
Furthermore, the limitations associated with the use of the app are not addressed. Specifically, Figures 3b, 4b, and 5b illustrate that the boxes created by the app do not always align with the anatomical region of interest. This limitation has not been discussed. It would be beneficial to include a discussion of this issue to provide a more comprehensive evaluation of the app’s accuracy and effectiveness.
Conclusions:
Line 431: I would suggest to change the term “disease” with “symptoms”.
Line 433: The value of 38.8% is not mentioned elsewhere in the results section. Could you please clarify how this percentage was determined? According to the data, 43 animals exhibited severe nasal atrophy and 41 shoved mild atrophy, making a total of 84 animals. This represents 69% of the 121 animals. Please provide further details, as the overall numbers appear misleading if not clearly presented in the results section. A revision of the result section is recommended to ensure accuracy and clarity.
Line 435: This sentence is unclear, particularly with the use of the word “load” (consider using “presence” instead) and the terms “this pathogen” (which refers to B. bronchiseptica). Please revise the sentence to enhance clarity.
Author Response
Dear editors,
Thank you for giving us the opportunity to revise our manuscript “Evaluation of the presence of Bordetella bronchisepticain the respiratory tract of pigs necropsied by respiratory symptoms, and its coinfection with other pathogens “
We made all changes suggested by the three referees.
The comments of all three reviewers were very helpful and we hope we are able to address them entirely to your satisfaction. Thank you very much for your time and help with reviewing and editing our manuscript.
We have prepared a revised version of the manuscript, highlighting the changes in red, and we have prepared a point to point response in order to answer the questions put by the reviewers.
The document has been revised by an English native speaker and the certificate of this revision is attached.
Reviewer 1
We highly appreciate the reviewer´s constructive and helpful efforts to improve the paper by his remarks. We take over all suggestions for re-phrasing and we eliminated the spelling mistakes detected by the reviewer. We have prepared a revised version of the manuscript, highlighting the changes in red. The document has been revised by an English native speaker and the certificate of this revision is attached. Corrections by the native speaker are not highlighted. The discussion was requested to be re-ordered and re-written, so that only new aspects are highlighted in right. Please find here our point-to-point responses:
- In general
Overall, the manuscript is moderately well-written but has structural deficits, particularly in the results and discussion sections. Although I am not a native English speaker, I noticed that certain parts of the text could benefit from improvement in terms of language. I would kindly suggest having the manuscript reviewed by a native English speaker to enhance its clarity and readability.
Answer: The manuscript has been revised by a native English speaker. The certificate is attached. The corrections of the English native speaker are not highlighted in red. We changed the discussion section in total as recommended. Only new aspects are highlighted in red. In the discussion we now included also a methodological criticism addressing the three major weak points of the study:
- Age of the pigs not known (Line 493-497)
- No histological examination of nasal tissue (Line 513-529)
- Diagnostic from nasal swabs only focussed on bronchiseptica and P. multocida and no other bacterial and viral pathogens (Line530-540).
- I would recommend including the use of the app in the title, as it represents the most innovative element of the article and, in my opinion, provides the most valuable results.
Answer: The title has been changed as suggested: Evaluation of porcine nose atrophy by automatic imaging and detection of Bordetella bronchiseptica (Line 1-3)
Abstract Line 38: I noticed a discrepancy between the abstract and the results section; specifically, the abstract reports 29.9%, while the results section (line 261) mentions 28.9%. Please adjust.
Answer: These were rounding errors, which had to be corrected. The numbers were corrected to 29.0% and adjusted (Line 33).
Line 72: Please change the term “pneumonic” to “pneumonia” or “pneumonic lesions” and add the term “conchae” or “turbinate” after “nasal”. These modifications ensure clarity and precision.
Answer: The terms were modified as suggested (Line 62).
Reference #7 does not describe necrosis, neutrophils, and hemorrhages. It appears that the sentence has been copied from reference #3, which discusses the dermonecrotic toxin of Bordetella causing turbinate atrophy and pneumonic lesions characterized by necrosis, hemorrhage, neutrophil accumulation, and fibrosis. Please verify the references and ensure that they are cited correctly.
Answer: Reference 7 was exchanged by a reference more appropriate (Sakano T.; Okada, M.; Taneda, A.; Ono, M.; Sato, S. Experimental atrophic rhinitis in 2 and 4 month old pigs infected se-quentially with Bordetella bronchiseptica and toxigenic type D Pasteurella multocida. Vet. Microbiol. 1992, 31, 197–206. https://doi.org/10.1016/0378-1135(92)90078-8) and the sentence was rephrased (Line 62-64).
Line 77: Please specify what 'older pigs' refers to in the context of the study. Additionally, the sentence currently lacks a clear subject. It would be helpful to include 'B. bronchiseptica' as the subject to clarify the statement.
Answer: The age-group was specified and the sentence was rephrased as suggested (Line 69-70)
Material and Methods
Line 129: In addition to body weight, I recommend including the age of the animals in the study. Since all the examined pigs exhibited respiratory symptoms, I would expect that some might show signs of wasting. Furthermore, the described body weight range suggests that an animal weighing 3.7 kg in this age group could be wasting. The animals are described as nursery pigs; however, based on my experience, I have not encountered a nursery pig weighing 42 kg. Please provide additional information or confirm whether this animal was accurately classified as a nursery pig.
Answer: We completely agree and deleted the statement, that pigs were nursery pigs. We added, that the age of the pigs could not be deduced from body weight, because most of the pigs might have wasted according to the pathomorphological findings. The weak point of missing age information is addressed in the discussion now (Line 129-133, 493-497).
Line 130: Please specify the symptoms, indicating whether they affect only the upper respiratory tract (e.g., sneezing, coughing) or the lower respiratory tract (e.g., dyspnea), or if both are involved.
Answer: The farmers always stated, that the pigs coughed when they were alive (the time before they delivered the died or euthanized pigs for necropsy). This observed clinical sign was added to the text (Line 126-127).
Figures 1b) – 5b): The photos in panel b) appear to be out of focus.
Answer: The original fotos created by the app were out of focus. We now improved the quality of the photos/chose photos with better quality (page 7).
Overall, the data are not clearly presented. It is confusing that the total number of animals varies, sometimes being reported as 122 and other times as 107. To enhance clarity, the sentence stating that 14 samples could not be cultured should be moved to the beginning of the paragraph rather than being placed at the end. Additionally, Figure 6 is not optimally designed; I recommend including the number of animals either next to or within the bars to improve clarity and provide a clearer representation of the data.
Answer. As suggested the information about the included animal numbers for evaluation is now given in the beginning of the paragraph (Line 312-320). Figure 6 is deleted and replaced by a new Table 3 providing the data more clearly and given additional explanation in the table legend (Line 324-328).
Line 271: Please add 'n=27' after 'lungs positive for B. bronchiseptica,' as this number is only visible in Figure 7 but not mentioned in the text. Additionally, specify the number of cultured pathogens (X/27; XX %) to make the information clearer and easier to read. It would also be beneficial to include the number of lungs with a monoinfection by B. bronchiseptica alone. According to my calculations, for 10 out of the 27 animals, it is unclear how many had E. coli cultured (as noted in Figure 7 but not discussed in the results or discussion sections), or if only B. bronchiseptica was cultured.
Answer: The number of lungs positive for B. bronchiseptica (n=27) was added, as well as the number of lungs with only detection of B. bronchiseptica (10/107) (Line 360-366). A paragraph about specific pathogens and pathogen combinations in different types of pneumonia is included (Line 331-358).
Line 283: Please clarify if B. bronchiseptica in the nose was detected by PCR or culture.
Answer: The information (by PCR) was added (Line 382-384)
Figure 11: The label “Figure 11” is out of order, as is “Figure 10”, which comes afterward. This minor typographical error should be corrected.
Answer: We are sorry for this mistake. The figure numbers are changed and Fig. 11 is now Fig. 9 (Line 401).
Line 303 – 305: As previously mentioned in “Line 129”, I suggest using the age of the animals as a more appropriate parameter for statistical analysis or considering both parameters (age and weight) for the same reasons previously discussed. Including results based on these parameters could provide further insights.
Answer: Due to the fact, that no information about the real age of the pigs was available, this is indicated in the material and methods part (Line 130-133), in the results part now (Line 310-311) and is now also discussed in the discussion section (Line 493-497). Former Fig. 10 is deleted.
Since the data are available, I would recommend including additional analysis and results. Specifically, it would be interesting to illustrate which pathogenic bacteria from the lungs were cultured alongside multiple pathogens, particularly to see if the “co-infection” consistently involves the same 2 or 3 pathogens.
Answer: Coinfections were evaluated in detail resulting in no clear pattern. Next to pigs, in which only one pathogen was detectable, in total 24 different coinfection patterns were found (not shown). Most often was the pattern “Gps and S.suis”, followed by “B.bronchiseptica and S.suis”. A summary of this evaluation is given in the results section also including the aspects of bacteria found in the different types of pneumonia (next comment) (Line 331-358).
Additionally, in relation to the pattern of pneumonia described under 2.1 Sample Collection, it would be valuable to investigate whether there is a correlation between these patterns and the cultured bacteria. It is known which pathogens can lead to specific pattern of pneumonia. And such a correlation could be done at least for APP (hemorrhagic necrotizing pneumonia) and GPS (fibrinous pleuritis).
Answer. As suggested, pattern of pneumonia were allocated to the pathogens, but no clear pathogen pattern was found. For APP mainly a catarrhalic purulent bronchopneumonia and a haemorrhagic pneumonia was found. For Gps nearly all pigs showed a catarrhalich-purulent bronchopneumonia. A summary of the evaluation is included in the results section now also unravelling coinfections of different pathogens found in lungs with different types of pneumona (see prior comment) (Line 331-358).
Discussion:
In general, the discussion could benefit from clearer structuring and a stronger focus on the results of this study. A more organized presentation would enhance the interpretation of the findings and their relevance.
We changed the discussion section in total, because all reviewers addressed missing points and a restructuring of the discussion was recommended. In the discussion we now included a methodological criticism addressing the three major weak points of the study:
- Age of the pigs not known (Line 493-497)
- No histological examination of nasal tissue (Line 513-529)
- Diagnostic from nasal swabs only focussed on bronchiseptica and P. multocida and no other bacterial and viral pathogens (Line530-540).
Line 352 - 353: This sentence is unclear, particularly with the use of the word “affecting”, which does not fit in the context. Please revise it to improve clarity.
Answer. The sentence has been revised as suggested (Line 456-458)
Line 353: Reference #45 Please verify the references and ensure that they are cited correctly.
Answer: The reference 45 was replaced by two more appropriate (Segura et al. 2016)
Line 355: Reference #46 Please verify the references and ensure that they are cited correctly. The article does not provide information on S. suis causing atrophic rhinitis.
Answer: Reference 46 was verified, the sentence was rephrased (Line 459-460) and a reference about S.suis nasal colonization was added (Hau et al. 2023).
Line 358 – 361: The information in these sentences could have been confirmed or excluded in this study through histological examination. Such an investigation would have been appropriate in the study design to provide further results and validation of the findings.
Answer: Histological examination as the gold standard for diagnostic of pathological changes was not performed in this study. This weak point is now included in the discussion section. Due to the fact, that histological findings in porcine nose and tonsils as interfaces to the surroundings containing large immune cell aggregates and being colonized by microbiota and potential pathogens are difficult to interpret in conventional pigs, we decided not to perform histological examination in this study. For diagnosis of cytomegalovirus this would have been helpful. These aspects are discussed now (Line 513-529).
Line 360: First use of an abbreviation should be written out in full terms. Please make the necessary changes.
Answer: The abbreviation is written in full term (Line 465-466)
Line 361: References #12,48,49: Please verify the references and ensure that they are cited correctly.
Answer: The references were verified. Reference 12 and 48 describe the high detection rates of G.parasuis in the respiratory tract of healthy pigs and pigs with respiratory disease. In reference 49 the process of nasal infection with G.parasuis is described resulting in suppurative rhinitis and epithelial cell degeneration. Coinfection with B.bronchiseptica might aggravate rhinitis (New Referencenumbers 46-48, Line 466).
Line 365: Mycoplasma hyorhinis has been renamed Mesomycoplasma hyorhinis. Please update the taxonomic classification.
Answer: M.hyorhinis was renamed as suggested (Line 469).
Line 373: Based on Reference #53, the cited article does not report that nasal mucosal inflammation is more common in young pigs. Please review the literature carefully and ensure that it is cited accurately.
Answer: The sentence was changed. We decided to rephrase the sentence with focus on the weaker nasal barrier function in young piglets (Line 509-512).
Line 385: As mentioned previously, establishing a correlation between the NLS and the weight of the animal is inappropriate. Consequently, any conclusions regarding the NLS not being age-related should be reconsidered.
Answer: The statement about any correlation between weight /age and NLS was skipped and the sentences was rephrased. The missing age data are critically discussed (Line 492-504).
Line 387: The information regarding the average weight of the study animals being 13 gg should be included in either the M&M or results section.
Answer: The average weight of the study pigs (15.6 kg) is included in the M&M section now (Line 129-130).
Line 399: P. multocida typographical error, should be written in italics.
Answer: The paragraph was rephrased and the typographical error was corrected (Line 475-479).
Furthermore, the limitations associated with the use of the app are not addressed. Specifically, Figures 3b, 4b, and 5b illustrate that the boxes created by the app do not always align with the anatomical region of interest. This limitation has not been discussed. It would be beneficial to include a discussion of this issue to provide a more comprehensive evaluation of the app’s accuracy and effectiveness.
Answer: The limitations of the app are further discussed now using the examples shown in figures 3 and 5, in which theboxes created by the app do not always correspond to the anatomical region of interest (Line 564-585).
Conclusions:
Line 431: I would suggest to change the term “disease” with “symptoms”.
Answer: The word was changed as suggested (line 590).
Line 433: The value of 38.8% is not mentioned elsewhere in the results section. Could you please clarify how this percentage was determined? According to the data, 43 animals exhibited severe nasal atrophy and 41 shoved mild atrophy, making a total of 84 animals. This represents 69% of the 121 animals. Please provide further details, as the overall numbers appear misleading if not clearly presented in the results section. A revision of the result section is recommended to ensure accuracy and clarity.
Answer: We excuse for the typing mistake (35.5% of animals showed NLS 5-18) and the misleading phrasing. We added more details about the pigs with low nasal lesion scores (NLS ≤4) in the results section (Line 393-396) and corrected the number in the conclusions (Line 591).
Line 435: This sentence is unclear, particularly with the use of the word “load” (consider using “presence” instead) and the terms “this pathogen” (which refers to B. bronchiseptica). Please revise the sentence to enhance clarity.
Answer: The sentence was rephrased as suggested (Line 593-594).
Reviewer 2 Report
Comments and Suggestions for Authors
Thank you for letting me review “Evaluation of the Presence of Bordetella bronchiseptica in the Respiratory Tract of Pigs Necropsied by Respiratory Symptoms, and Its Coinfection with Other Pathogens”
I find the work behind well designed, explained and performed as well as the paper well written.
I have very few comments only and they are mainly typo’s .
#####
Title: Please put all of” Bordetella bronchiseptica“ in italics.
L. 54-56: This is a long sentence, and a bit hard to understand. I suggest you rephrase a bit.
L. 73 + 115: B.-bronchiseptica “ please remove the ‘-‘ in B. bronchiseptica. Also, for P.-multocida (L115) and A.-pleuropneumoniae (L 182). Please go through the full manuscript to change this.
L. 77-79 : “In older pigs, can be an…” A word is missing after pigs? Please rephrase.
L. 129: Please state which year the pigs were sent for necropsy
L. 145 + 123: AI DIAGNOS : Please state a reference for this program. The AI training is well described later in L 234-241.
L. 146: Do you by removal of the “facial mask” mean the skin of the face? Please elaborate a bit.
L. 220: A general question: is there an NLS value which correlates to atrophic rhinitis? Perhaps elaborate in the discussion. Did any of the herds or pigs suffer from clinical atrophic rhinitis. Do we know the clinical impact of mild and severe NLS?:
L. 229: Table 2. Please write out NLS. Please arrange the table according to the guidelines of Animals: ie no visible vertical lines (similar to Table 1)
L. 257: Please state the number of herds the 121 pigs originate from.
L.275: Figure 7 For ease of intuitively reading the figure, I would suggest placing the number of positive pigs above the columns “n=27”. The same comment for Figure 9.
L.307: Figure 10. Please Write out NLS. In general remember that Tables and Figures should be self-explanatory. The same comment for Figure 12.
L. 297: Figure 11 comes before Figure 10. Please correct the order (or numbers) of the figures.
L. 314: To me it seems that the automated NLS on average scores a bit higher than the visual score
L. 417: Please reflect a bit on the Spearman-Rank-Correlation-Coefficient of 0.61 which is not considered a high but a moderate correlation.
Author Response
Dear editors,
Thank you for giving us the opportunity to revise our manuscript “Evaluation of the presence of Bordetella bronchisepticain the respiratory tract of pigs necropsied by respiratory symptoms, and its coinfection with other pathogens “
We made all changes suggested by the three referees.
The comments of all three reviewers were very helpful and we hope we are able to address them entirely to your satisfaction. Thank you very much for your time and help with reviewing and editing our manuscript.
We have prepared a revised version of the manuscript, highlighting the changes in red, and we have prepared a point to point response in order to answer the questions put by the reviewers.
The document has been revised by an English native speaker and the certificate of this revision is attached.
Reviewer 2
We highly appreciate the reviewer´s constructive and helpful corrections, which will improve the paper. We take over all suggestions for re-phrasing and we eliminated the mistakes detected by the reviewer. We have prepared a revised version of the manuscript, highlighting the changes in red. The document has been revised by an English native speaker and the certificate of this revision is attached. Corrections by the native speaker are not highlighted. The discussion was requested to be re-ordered and re-written, so that only new aspects are highlighted in right.
In the discussion we now included a methodological criticism addressing the three major weak points of the study:
- Age of the pigs not known (Line 493-497)
- No histological examination of nasal tissue (Line 513-529)
- Diagnostic from nasal swabs only focussed on B. bronchiseptica and P. multocida and no other bacterial and viral pathogens (Line 530-540).
Please find here our point-to-point responses:
Title: Please put all of” Bordetella bronchiseptica“ in italics.
Answer: The titel was rephrased according to reviewer suggestions and the bacterial species name is written in italics now (Line 3)
- 54-56: This is a long sentence, and a bit hard to understand. I suggest you rephrase a bit.
Answer: The paragraph was rephrased as suggested (Line 43-48).
- 73 + 115: B.-bronchiseptica “ please remove the ‘-‘ in B. bronchiseptica. Also, for P.-multocida (L115) and A.-pleuropneumoniae (L 182). Please go through the full manuscript to change this.
Answer: The hyphens has been deleted within the bacterial names allover the manuscript.
- 77-79 : “In older pigs, can be an…” A word is missing after pigs? Please rephrase.
Answer: The sentence has been rephrased (Line 69).
- 129: Please state which year the pigs were sent for necropsy
Answer: We have to clarify the sampling time. Due to the fact, that sample size was increased in an additional time period, sampling took place within eleven month, but in two different periods: between August 2022-January 2023 as well as between October 2023 and February 2024. This information is added now (Line 125-126).
- 145 + 123: AI DIAGNOS : Please state a reference for this program. The AI training is well described later in L 234-241.
Answer: Unfortunately no peer review papers exist so far, because the app is a prototype. The app “AI Diagnos” was published in conference papers. One reference of a conference paper published at the International Pig Veterinary Society Congress in 2022 is included in the text and in the reference list now (Jorda et al. 2022).
- 146: Do you by removal of the “facial mask” mean the skin of the face? Please elaborate a bit.
Answer: The sentence is rephrased as suggested and the process of skin removal is described in more detail (Line 142-145).
- 220: A general question: is there an NLS value which correlates to atrophic rhinitis? Perhaps elaborate in the discussion. Did any of the herds or pigs suffer from clinical atrophic rhinitis. Do we know the clinical impact of mild and severe NLS?:
Answer: In another study using the same scoring scheme, nasal atrophy was related to weight and daily weight gain. Pigs with moderate NLS (5-9) and severe NLS (10-18) showed a reduced growth performance and lower slaughter weights (Donko et al. 2005). In contrast to our study, authors examined pigs infected by B. bronchiseptica and toxigenic Pasteurella multocida strains, which means that pigs suffered from progressive atrophic rhinitis. In contrast to that examined pigs in our study showed non-progressive atrophic rhinitis, because P. multocida strains were negative for the Pasteurella multocida toxin. This aspect and the respective reference (Donko et al. 2005) is now included in the discussion section (Line 475-479).
The selected cut off of a nasal lesion score of 5 for severe nasal lesions is shortly explained in the M&M section (Line 242-243).
- 229: Table 2. Please write out NLS. Please arrange the table according to the guidelines of Animals: ie no visible vertical lines (similar to Table 1)
Answer: The table heading has been adapted and the table format was changed as suggested (Line 257-260).
- 257: Please state the number of herds the 121 pigs originate from
Answer: The information is added. Piglets originated from 64 farms and in total pigs from 39 farms were positive for B.bronchiseptica in the nose by PCR (Line 315-316).
L.275: Figure 7 For ease of intuitively reading the figure, I would suggest placing the number of positive pigs above the columns “n=27”. The same comment for Figure 9.
Answer: Fig. 7 and 9 were changed according to the suggestion and are now renamed (Fig 6 and 8)
L.307: Figure 10. Please Write out NLS. In general remember that Tables and Figures should be self-explanatory. The same comment for Figure 12.
Answer: Fig. 10 was deleted, because a relation between body weight and nasal lesion score is not valid because the age is lacking and pigs had been wasting. This has also adressed by the other reviewers and is now critically discussed.
In Fig. 12, which is now Figure 10, adaptations have been made as suggested.
- 297: Figure 11 comes before Figure 10. Please correct the order (or numbers) of the figures.
Answer: We are sorry for this mistake. The figure numbers are changed because finally two figures have been deleted.
- 314: To me it seems that the automated NLS on average scores a bit higher than the visual score
Answer: The automated score is not higher than the visual score, because both were in visual median 4, range 0-15 and automated in median 3, range 0-12. These descriptional findings are included in the results section (Line 412-413).
- 417: Please reflect a bit on the Spearman-Rank-Correlation-Coefficient of 0.61 which is not considered a high but a moderate correlation.
Answer: The found moderate (to high) Spearman-Rank-Correlation-Coeffficient is addressed in the discussion as suggested (Line 576-582).
Reviewer 3 Report
Comments and Suggestions for Authors
Interesting but...
1. Need to clearly define group - nursery - these are not piglets less than 7kg and more than 30kg - they must be clearly defined where on the farm these pigs came from.
2. The study only looked at bacteria and this needs to be clear
3 No normal pigs were included need to clearly define normal nasal antomy and pathology = especially role of inclusion body rhinitis
4.Insufficient bacteriology of nasal cavity. No bacteriology of the tonsils?
5. Were any farms negative to B. bronchiseptica? Commensal.

Author Response
Dear editors,
Thank you for giving us the opportunity to revise our manuscript “Evaluation of the presence of Bordetella bronchisepticain the respiratory tract of pigs necropsied by respiratory symptoms, and its coinfection with other pathogens “
We made all changes suggested by the three referees.
The comments of all three reviewers were very helpful and we hope we are able to address them entirely to your satisfaction. Thank you very much for your time and help with reviewing and editing our manuscript.
We have prepared a revised version of the manuscript, highlighting the changes in red, and we have prepared a point to point response in order to answer the questions put by the reviewers.
The document has been revised by an English native speaker and the certificate of this revision is attached.
Reviewer 3
We highly appreciate the reviewer´s constructive and helpful corrections, which will improve the paper. We have prepared a revised version of the manuscript, highlighting the changes in red. The document has been revised by an English native speaker and the certificate of this revision is attached. Corrections by the native speaker are not highlighted. The discussion was requested to be re-ordered and re-written, so that only new aspects are highlighted in right.
In the discussion we now included a methodological criticism addressing the three major weak points of the study:
- Age of the pigs not known (Line 493-497)
- No histological examination of nasal tissue (Line 513-529)
- Diagnostic from nasal swabs only focussed on B. bronchiseptica and P. multocida and no other bacterial and viral pathogens (Line 530-540).
Please find here our point-to-point responses:
Need to clearly define group - nursery - these are not piglets less than 7kg and more than 30kg - they must be clearly defined where on the farm these pigs came from.
Answer: We completely agree. No age information is available for the pigs sent for necropsy. Farmers stated, that they are from the nursery unit, but according to the lung alterations, they might have been wasted, so that age cannot be deduced from weight. For this reason we deleted the Figure 10 showing the relation between weight and NLS.
It is not seldom, that in some farms with short suckling period (21 days in average, but sometimes piglets born late can also be in the age of less than 17 days) weaning weights are much lower than 7 kg.
The disadvantage of lacking information about age is now included in the discussion section (Line 493-497). The Material and Methods Section was adapted rephrasing the age-group description (Line 130-133).
The study only looked at bacteria and this needs to be clear
Answer: Now already in the abstract the restriction of diagnostic from nasal swabs to B. bronchiseptica and P. multocidais clearly mentioned (Line 29-31), as well as in the M&M section (Line 157-159). In addition, this point is included in the discussion as one major weak point (Line 530-540).
No normal pigs were included need to clearly define normal nasal antomy and pathology = especially role of inclusion body rhinitis
Answer: The missing histological examination and therefore the possibility of not diagnosing cytomegalovirus is addressed in the discussion now (Line 513-529). Normal nasal anatomy has been published in detail and further references about the nasal scoring and the justification for the NLS cut off of ≥5 for NPAR are included now (Donkó et al. 2005, Ref. 34).
Insufficient bacteriology of nasal cavity. No bacteriology of the tonsils?
Answer: This point is addressed as one major weak point in the discussion now (Line 530-540). Interpretation of pathogens form mucosal surfaces as interface to the surroundings as tonsil and nose (air filter organ) are difficult, because nearly all pigs are colonized with pathogens potentially causing rhinitis. According to detection rates usually no difference between healthy and diseased pigs is found.
Were any farms negative to B. bronchiseptica? Commensal.
Answer: In total pigs from 64 farms were examined and pigs from 39 farms were tested positive for B. bronchiseptica by PCR in the nose. This means that 39% of the farms were negative in this study. This information is included in the results section (Line 315-316).
The sample size (pigs examined per farm) is not adequate to prove, that a farm is really negative. Due to the fact, that B.bronchispetica is a colonizer of the nasal cavity in healthy pigs, it can be expected, that all farms are positive. This point is addressed in the discussion (Line 435-442).
Remarks and requested revisions in the manuscript
Answer: All remarks written directly in the manuscript were addressed at the respective positions and highlighted in red in the manuscript.
Simple summary:
Answers to open questions are implemented:
-The first sentence is changed (Line 12-15)
-an age range is mentioned (pigs originating form nursery and beginning of fattening) (Line 12)
-No toxigenic P. multocida were found in pigs of this study, so that only NPAR was diagnosed (Line 18-19)
Introduction:
The first paragraph about Porcine respiratory disease complex is shortened and rephrased as suggested (Line 43-48)
-we had normal (conventional) pigs included in the study, which show respiratory signs. We exspect, that we would have detected atrophic rhinitis due to sensitive methods, if it would have been there (toxigenic P. multocida) (Line 320-321). In Germany the prevalence of PAR is very low. In the last 5 years we had not a single case in our diagnostic lab.
We think, that the examined pig group is appropriate for diagnostic of PAR, because recommendations in the official national monitoring programs is to sample nasal swabs in pigs of approx. 5-8 weeks of age.
Material and method
-The first paragraph is rewritten (Line 124-133). Unfortunately the correct age of pigs was not known. For that reason the relation between weight and nasal lesion score was deleted. We decided to leave pigs in the study (also those weighing above 30 kg), because all pigs come from nursery or beginning of fattening independent of weight. The wording was corrected throughout the manuscript. One focus is on the scoring procedure of the anatomical sites. For sure environments were all different between the farms. Also the nursery units on the different farms are not comparable due to different environmental influences. These environmental factors cannot be addressed in this study.
-The blurred figures 1-5 were exchanged. A picture with NLS=0 is included (Fig. 1)
Results:
-The confusing figure 6 is deleted and an overview of findings in pigs is summarized in a new Table 3 (Line 323-325)
-nasal swabs were only analysed for B.bronchiseptica and P. multocida, which is one weak point of the study as discussed (Line 530-540). Nevertheless also in healthy pigs other colonizing bacteria can be found in the nose and interpretation of findings will be difficult.
-Figure 9 as new Figure 8 was revised as suggested
-The old figure 10 is deleted, because the relation between age (not known) and weight is not clear, so that a relation between weight an NLS is not meaningful.
Discussion:
-The lack of histological examination and diagnostic of inclusion bodies/cytomegalovirus is one weak point of the study, which is included in the discussion now (Line 513-529).
Round 2
Reviewer 1 Report
Comments and Suggestions for Authors
All of my remarks and comments have been comprensively addressed and improved by the authors. They have investited considerable effort into refining this manuscript, resulting in a highly commendable outcome. I appreciate the care and attention they have dedicated to implementig the necessary revisions.
There remains only one minor issue in line 52 (Mycoplasma hyopneumoniae has been renamed Mesomycoplasma hyopneumoniae), aside from this, the manuscript is in excellent condition.
